# Assessment of the Potential Skin Application of *Plectranthus ecklonii* Benth.

**DOI:** 10.3390/ph13060120

**Published:** 2020-06-10

**Authors:** Marisa Nicolai, Joana Mota, Ana S. Fernandes, Filipe Pereira, Paula Pereira, Catarina P. Reis, Maria Valéria Robles Velasco, André Rolim Baby, Catarina Rosado, Patrícia Rijo

**Affiliations:** 1CBIOS—Universidade Lusófona Research Centre for Biosciences & Health Technologies, Campo Grande 376, 1749-024 Lisbon, Portugal; marisa.nicolai@ulusofona.pt (M.N.); joana.mota@ulusofona.pt (J.M.); ana.fernandes@ulusofona.pt (A.S.F.); pereira.filipepereira@gmail.com (F.P.); paula.pereira@ulusofona.pt (P.P.); catarina.rosado@ulusofona.pt (C.R.); 2CERENA—Centre for Natural Resources and the Environment, Instituto Superior Técnico (IST), Universidade de Lisboa, Av. Rovisco Pais, 1049-001 Lisbon, Portugal; 3iMed.ULisboa Research Institute for Medicines and Pharmaceutical Sciences, Universidade de Lisboa—Faculdade de Farmácia, Av. Prof. Gama Pinto, 1649-003 Lisbon, Portugal; catarinareis@ff.ulisboa.pt; 4IBEB, Faculdade de Ciências, Universidade de Lisboa, 1749-016 Lisbon, Portugal; 5Department of Pharmacy, School of Pharmaceutical Sciences, University of São Paulo BEB, 580 Lineu Prestes Av., Bloco 15, São Paulo/SP 05508-900, Brazil; mariavaleriavelasco@usp.br (M.V.R.V.); andrerb@usp.br (A.R.B.)

**Keywords:** *Plectranthus ecklonii*, parvifloron D, rosmarinic acid, in vitro permeation, sun protection factor, UV protection, antibacterial

## Abstract

*Plectranthus ecklonii* Benth. has widespread ethnobotanical use in African folk medicine for its medicinal properties in skin conditions. In this study, two different basic formulations containing *P. ecklonii* extracts were prepared, one in an organic solvent and the other using water. The aqueous extract only contained rosmarinic acid (RA) at 2.02 mM, and the organic extract contained RA and parvifloron D at 0.29 and 3.13 mM, respectively. RA in aqueous solution permeated skin; however, in *P. ecklonii* organic extract, this was not detected. Thus, *P. ecklonii* aqueous extract was further studied and combined with benzophenone-4, which elevated the sun protection factor (SPF) by 19.49%. No significant cytotoxic effects were observed from the aqueous extract. The *Staphylococcus epidermidis* strain was used to determine a minimum inhibitory concentration (MIC) value of 10 µg·mL^−1^. The aqueous extract inhibited the activity of acetylcholinesterase by 59.14 ± 4.97%, and the IC_50_ value was 12.9 µg·mL^−1^. The association of the *P. ecklonii* extract with a UV filter substantially elevated its SPF efficacy. Following the multiple bioactivities of the extract and its active substances, a finished product could be claimed as a multifunctional cosmeceutical with broad skin valuable effects, from UV protection to antiaging action.

## 1. Introduction

The use of plants for medicinal purposes has been continuous throughout human history. Nowadays, there is an increasing academic, commercial, regulatory, and public interest in medicinal plants, and in the last decades, there has been a sharp increase in the inclusion of plant extracts and essential oils in health products. This tendency can be explained, amongst other factors, by the satisfaction that consumers experience when buying “naturals”, since they are perceived as being safer than “synthetic” ingredients [1].

*Plectranthus ecklonii* Benth. is a shrub widely distributed in South Africa, Australia, and Northern and Southern America. *Plectranthus* species (Lamiaceae) contain several antioxidant compounds and exhibit more than a few effects such as antimicrobial, antifungal, and anti-inflammatory activities. It is used as an ethnobotanical plant in South Africa for treating nausea, vomiting, stomach aches, and respiratory problems. The native people of Zimbabwe used aerial parts of *P. ecklonii* for the treatment of skin issues and hyperpigmentation problems [2]. To fully understand the properties and advantages of the use of topical formulations containing *P. ecklonii* extracts, a systematic characterization of its main compounds and the determination of their biological activity is essential.

Polyphenols, reported as chemopreventive agents, can inhibit and prevent damage by UV exposure that can cause skin disorders, promoting oxidative stress, inflammation, and DNA damage [3,4,5].

The main phenolic compound of *P. ecklonii* is rosmarinic acid (RA, Extrasynthese, Genay, France) (Figure 1A), a common compound in the Lamiaceae family that exhibits many bioactivities, mainly antioxidant and anti-inflammatory [2]. Photoprotective and melanogenic properties have also been described [6]. On the other hand, the major abietane diterpenoid of *P. ecklonii*, parvifloron D (ParvD, previously isolated [7]) (Figure 1B), presents antibacterial activity, which can be associated with the folk uses of the plant for treating gastrointestinal distress and skin diseases [8]. Additionally, antityrosinase activity [9] and cytotoxicity properties [7,10] have been reported for ParvD.

To obtain the optimal benefits of topical formulations, the compounds that are applied over cutaneous tissue must reach their action sites to perform their biological activities. Compounds that act on deeper skin layers have to penetrate the stratum corneum to reach the cells localized in the lower strata of the epidermis, the dermis, or even the cutaneous microcirculation. On the other hand, cosmetic formulations are generally considered safer if the permeation of their main compounds is limited to the upper layers of the skin and does not reach the microcirculation. Thus, in the assessment of a potential candidate for topical administration, skin permeation assays are indispensable [8,9].

In this work, we investigated the biological activity of two different extracts of *P. ecklonii* and tested the permeation of the main components of the plant extracts through human skin. The ethanol/propylene glycol (PG, Sigma-Aldrich, St. Louis, MO, USA) (70:30, *v*/*v*) extract contains RA and ParvD, the last as the major compound, and the aqueous extract contains only RA in a higher concentration than that found in the ethanol/PG extract. The inclusion of an extract containing a cytotoxic compound in a formulation for topical application seems unsafe and risky. Therefore, it was decided to continue the study only with the aqueous extract that showed low cytotoxicity effects. Additionally, the potential of *P. ecklonii* extract in terms of antioxidant, antiacetylcholinesterase, and antimicrobial activities was assessed.

## 2. Results and Discussion

*P. ecklonii* has interesting biological and cosmetic applications, especially in African folk medicine. The scientific validation study of the potential cosmetic and pharmacological action of this plant is important for additional economic and pharmaceutical value. For skin proposes, we prepared two extracts using *P. ecklonii*, one aqueous and another using 70% ethanol and 30% PG (*v*/*v*). We studied these potential cosmetic extracts considering previous studies [7,9,11] for their chemical composition and biological activities. It is well known that *Plectranthus* spp. has a high quantity of the bioactive RA [12] but also contains the cytotoxic ParvD [7]. Therefore, it is very important to evaluate and quantify these constituents for the efficacy and safety assessment of these potential skin application extracts.

### 2.1. In Vitro Permeation Studies

To start the permeation study, RA and ParvD were identified and quantified in plant extracts by HPLC-DAD using analytical curves with ranges of 0.05–2.00 mM for RA (*y* = 4758.3*x* − 144.4, *R*^2^ = 0.993) and 0.06–3.50 mM for ParvD (*y* = 8009.3*x* − 500.3, *R*^2^ = 0.999). The limits of detection (LODs) for RA and ParvD were 0.01 and 0.01 mM, respectively, and the limits of quantification (LOQs) for RA and ParvD were 0.05 and 0.01 mM, respectively. Chromatograms and UV-visible spectra of RA standard, parvifloron D, *P. ecklonii* extracts in water, and ethanol/PG (70:30, *v*/*v*) are shown in Figure 2.

The initial concentrations of all the solutions used in the permeation studies are indicated in Table 1. Aliquots collected at 4, 8, 24, and 48 h after the start of the permeation assay and residues were resuspended in methanol after drug extraction from the skin, then they were all analyzed with a HPLC-diode array detector (DAD, Santa Clara, CA, USA) Agilent Technologies 1200 Infinity Series system (Santa Clara, CA, USA) with the ChemStation software in combination with a LiChrospher^®^ 100, RP-18 (5 µm) Merck column (Darmstadt, Germany).

The permeation data after 48 h are summarized in Table 2. At this time point, 4% of the RA from the aqueous solutions permeated through the human skin. After 48 h, ParvD in ethanol/PG (70:30, *v/v*) and the corresponding ethanolic extract and the RA from the aqueous extracts were not detected in the receptor compartment. RA from the aqueous extract did not permeate the skin, maybe because other components inhibited or hindered the diffusion of the RA. The constituents of the extract probably modified the solubility and percent saturation of this compound so that the flux was lower from the extract because it had lower thermodynamic activity. On the other hand, ParvD in the solutions and the extract did not permeate the skin, maybe due to its extreme lipophilicity. The results seem to indicate that the multiple compounds from the *P*. *ecklonii* extract in water and ethanol/PG (70:30, *v/v*) (Figure 2) did not act as permeation enhancers for RA and ParvD.

On the other hand, after 48 h, the method applied to the epidermal membranes to extract the trapped compounds in the skin, at the end of the permeation study, showed that no compound was detected under these extraction and quantification conditions, since none of the samples contained RA or parvifloron D. These results showed that RA and parvifloron D do not permeate the epidermis in detectable amounts, which suggests that these phytochemicals may exert their antioxidant/free radical scavenging activity mainly at the cutaneous surface. Since cosmetic formulations should ideally refrain from crossing the skin barrier, this property is desirable from a safety perspective.

#### In Vitro Photoprotective Efficacy

In the studies to determine the SPF, it was noticed that *P. ecklonii* extract significantly improved the photoprotective effectiveness of benzophenone-4, even in aqueous solution. The bioactive extract association with the UV filter provided an SPF augmentation of 19.49% (ANOVA followed by Tukey test, *p* < 0.05).

Psotova and co-workers [11] reported that RA was able to reduce UVA damage to human keratinocytes (HaCaT) cultures through cell viability establishment by neutral red retention and by the level of extracellular lactate dehydrogenase (LDH, Sigma-Aldrich, St. Louis, MO, USA). In the study conducted by Sánchez-Campillo and coworkers [6], RA was orally administered and evaluated as a photoprotective agent in albino mice. They found that RA acted as a protector against UVA-caused lesions, since the animals treated with RA presented only slight skin dysplasia in comparison with the non-treated group. It is noteworthy to mention that UVA exposure generates reactive oxygen species (ROS) in the skin that induce tissue photodamage and photoaging, leading to cellular damage and cell death by apoptosis or necrosis. One strategy to overcome those kinds of UV injuries is by using photoprotection, preferentially, that developed with UV filters associated with compounds that exert scavenging and quenching activities against ROS [13,14,15]. The *P. ecklonii* extract containing RA has such properties, and beyond its ability to protect the skin from UVA damage, it also absorbs UVB radiation, which, consequently, increases the sunscreen’s SPF, as herein estimated in vitro.

### 2.2. Cytotoxicity Profile

The data obtained in the cytotoxicity evaluation of the extract are depicted in Figure 3. Under our experimental conditions, no considerable cytotoxic effects were observed. Only decreases in cell viability lower than 10% were observed for all the concentrations tested. Conversely, the positive control (H_2_O_2_ 10 mM) reduced cell viability to 1.38 ± 0.18% (data are not shown).

### 2.3. Antimicrobial Activity

The antimicrobial activity of the aqueous extract was screened against Gram-positive (*Staphylococcus aureus*, *S. epidermidis,* and *Enterococcus faecalis*) and Gram-negative bacteria (*Escherichia coli* and *Pseudomonas aeruginosa*) using the well diffusion test (data not shown). The positive results obtained against *S. epidermidis* were further studied, and the MIC value was 10 µg·mL^−1^. These results show that this extract is active in bacteria, which are part of the normal human flora and typically the skin flora. These antimicrobial results are in agreement with other *Plectranthus* spp. extract studies [9,12], which encourage the use of these extracts as new ingredients for skin formulation.

### 2.4. Acetylcholinesterase Inhibition Assay

The *P. ecklonii* aqueous extract inhibited the activity of acetylcholinesterase by 59.14 ± 4.97% when using 0.1 mg of extract per milliliter of the test solution. This result is in agreement with other values established for the inhibition of this enzyme with other *Plectranthus* extracts [12].

### 2.5. Antioxidant Activity

The antioxidant activity of *P. ecklonii* extract, regarding its ability to scavenge free radicals, was determined, and an IC_50_ value of 12.9 µg·mL^−1^ was found. This result was expected, comparing to that for other *Plectranthus* spp. studied for cosmetic proposes [12]. The interesting antioxidant activity is also a benefit that presents these extracts as potential new ingredients for skin formulations.

All these biological activities described together corroborate the cosmetic potential of these extracts for skin formulations.

## 3. Material and Methods

### 3.1. Plant Material

*P. ecklonii* Benth. was grown in the Hortum of the Faculty of Pharmacy (Latitude: 38°46′ N; Longitude: 9°08′ W; Altitude: 100 m; Lisbon University, Lisbon, Portugal) from seeds provided by the Herbarium of the National Botanic Garden of Kirstenbosch, South Africa. Entire plants were collected in July 1998, identified, and deposited in the Herbarium of the Instituto de Investigação Científica Tropical, Lisbon (ref. C. Marques S/No. LISC).

### 3.2. Chemicals

Absolute ethanol and sodium dihydrogen phosphate dihydrate extra pure were obtained from Riedel-de Haën (Seelze, Germany). Muller–Hilton medium was purchased from Biokar Diagnostic (France). Acetone, *N*-hexane, ethyl acetate, methanol, acetonitrile, trichloroacetic acid, and dimethyl sulfoxide were purchased from Merck (Darmstadt, Germany). Acetylcholinesterase, Ellman reagent, acetylthiocholine iodide, 5,5’-dithiobis-(2-nitrobenzoic acid) (DTNB), HEPES buffer, tacrine, 2,2-diphenyl-l-picrylhydrazyl, Dulbecco’s Modified Eagle’s medium (DMEM), fetal bovine serum, penicillin–streptomycin solution, hydrogen peroxide, trypsin, phosphate buffer saline, thiazolyl blue tetrazolium bromide, rosmarinic acid (RA), propylene glycol, and sodium phosphate dibasic dehydrate were obtained from Sigma-Aldrich (Darmstadt, Germany). Sodium chloride was supplied by José M. Vaz Pereira (Lisbon, Portugal). Benzophenone-4 was purchased from Mapric (São Paulo, Brazil). All other chemicals were of HPLC analytical grade, and they were used as received. Parvifloron D (ParvD) was isolated according to the method described by [16], and its NMR spectroscopic characterization is provided in the Appendix A.

### 3.3. Extract Preparation

The aqueous extract was obtained adding 100 mL of bi-distilled water to 10 g of ground fresh plant material, as previously described by Rijo and coworkers [12]. The mixtures were subject to microwave-assisted extraction for 3 min, under continuous irradiation of 2.45 GHz. The ethanol at 70% and PG at 30% (*v*/*v*) plant extract was prepared using 25 mL of the vehicle and 10 g of fresh plant material, which were stirred for 1 h at room temperature [8]. Both extracts were filtered through Whatman paper, separated into aliquots (1 mL), and frozen at −20 °C.

### 3.4. Permeation Experiments

#### 3.4.1. Human Skin Preparation

Tissue from human abdominal skin obtained after cosmetic surgery and following informed consent was used as epidermal membranes. Ethical approval was provided by the Ethics Committee of the School of Sciences and Health Technologies of the Lusófona University. After the removal of the adipose tissue by blunt dissection, the epidermis was separated by immersing the skin in a water bath at 60 °C for 1 min [17]. It was then fixed on a corkboard. The epidermis was carefully separated from the dermis, adapted on the filter paper, and stored at −20 °C. To perform the diffusion experiment, the epidermis was defrosted and cropped to an appropriate size.

#### 3.4.2. In Vitro Permeation Studies

Glass Franz-type vertical diffusion cells with a receptor chamber of 4 mL and a diffusional area of 0.95 cm^2^ were used to conduct the permeation in the epidermal membranes (*n* = 3). In the receptor compartment, isotonic phosphate-buffered saline (PBS, pH 7.4) was continuously stirred [8,18] and maintained in a water bath thermostatically controlled at 37 °C. A fixed volume (400 µL) of aqueous extracts, 70% ethanol and 30% PG (*v*/*v*) [6] extracts, or solutions of RA or ParvD were placed in each donor compartment. Aliquots of the samples were collected at 4, 8, 24, and 48 h.

To extract and measure the drug content retained on the skin surface, at the end of the 48 h permeation study, the skin tissue was rinsed 3 times with bi-distilled water for a total of 30 s and wiped off with an alcohol pad. The entire piece of skin was blotted dry with a paper towel. The skin was rinsed an additional time with bi-distilled water and blotted dry again. In an attempt to extract the drug, the skin was introduced into 10 mL of acetonitrile and shaken for 48 h at room temperature. After solvent evaporation, the residue was resuspended in methanol (400 µL) and once again analyzed with an HPLC-DAD.

#### 3.4.3. In Vitro Sun Protection Factor

The in vitro Sun Protection Factor (SPF) was estimated by UV spectrophotometry. Samples were evaluated in an aqueous vehicle containing the *P. ecklonii* extract uniquely at 10.0% (*w*/*v*) or in a binary association with benzophenone-4 at 6.0% (*w*/*v*). Samples were diluted in absolute ethanol to 0.2 mg·mL^−1^, and they were run from 290 to 320 nm using a 1.0 cm path length quartz cuvette (*n* = 3). Absolute ethanol was also used as a blank. SPF values were calculated with Equation (1) [19]. The results were statistically analyzed by ANOVA followed by a Tukey test, *p* < 0.05.
(1)$$\mathrm{SPF}=\mathrm{CF}\;\times \;\sum_{290\;nm}^{320\;nm}\times \;\mathrm{EE}(\lambda )\;\times \;I(\lambda )\;\times \;A(\lambda )$$

Equation (1) estimates the sun protection factor (SPF), where CF is the correction factor equal to 10, EE(λ) is the erythemal efficiency spectrum, I(λ) is the solar simulator spectrum as measured with a calibrated spectroradiometer, and A(λ) is the sample absorbance.

### 3.5. HPLC Analysis

RA and ParvD were detected and quantified with the HPLC-DAD Agilent Technologies 1200 Infinity Series system with the ChemStation software in combination with a LiChrospher^®^ 100, RP-18 (5 µm) Merck column. The samples were analyzed, injecting 20 µL and using a 1 mL·min^−1^ flow rate and a gradient composed of solution A (methanol), solution B (acetonitrile), and solution C (0.3% trichloroacetic acid in water) as follows: 0 min, 15% A, 5% B and 80% C; 20 min, 80% A, 10% B, and 10% C; and 28 min, 80% A, 10% B and 10% C. The standard methanol solutions were run under the same conditions, and the detection was carried out at 270 nm with a diode array detector (DAD). All the analyses were performed in triplicate. The limit of detection (LOD = 3 × blank standard deviation/scope linear regression) and limit of quantification (LOQ = 10 × blank standard deviation) for RA and ParvD were evaluated.

### 3.6. Cytotoxicity Profile

HaCaT keratinocytes were cultured in DMEM supplemented with 10% fetal bovine serum, 100 U·mL^−1^ penicillin, and 0.1 mg·mL^−1^ streptomycin. The cells were kept at 37 °C, under an atmosphere containing 5% CO_2_ in the air. The cytotoxicity profile of the aqueous extract was evaluated by the MTT assay. Briefly, the cells were seeded in 200 μL of culture medium per well in 96-well plates and incubated for 24 h. Afterward, cells were exposed to increasing concentrations of the extract up to 500 μg·mL^−1^ for 24 h. H_2_O_2_ (10 mM) was used as a positive control. The MTT assay was then carried out as previously described [20].

### 3.7. Antibacterial Activity

#### 3.7.1. Bacterial Strains

The in vitro antimicrobial study was carried out against five bacterial strains obtained from American Type Culture Collection (ATCC): three Gram-positive (*Staphylococcus aureus* ATCC 25923, *S. epidermidis* ATCC 12228, and *Enterococcus faecalis* ATCC 29212) and two Gram-negative bacteria (*Escherichia coli* ATCC 25922 and *Pseudomonas aeruginosa* ATCC 27853).

#### 3.7.2. Well Diffusion Assay

The well diffusion assay was used to screen the antibacterial activity of the aqueous extract. Mueller–Hinton culture medium (20 mL), placed in Petri dishes, was inoculated with 0.1 mL of a bacterial cell suspension matching a 0.5 McFarland standard solution. The suspension was homogeneously spread using a sterile swab over the medium surface. Wells with a 5 mm diameter were made in the agar plates, and 50 µL of 1 mg·mL^−1^ extracts reconstituted in DMSO was added into the wells. DMSO was used as the negative control, while 1 mg·mL^−1^ vancomycin and 1 mg·mL^−1^ norfloxacin solutions were used as positive controls for the Gram-positive and Gram-negative bacteria, respectively. After 24 h of incubation at 37 °C, the antibacterial activity was assayed through the measurement of the diameter, in mm, of the inhibition zone formed around the wells. All the samples were assayed at least in duplicate.

#### 3.7.3. Minimum Inhibitory Concentration (MIC) Evaluation

The minimum inhibitory concentration (MIC) value of the aqueous extract was evaluated by the microdilution method. One hundred microliters of Mueller–Hilton medium was placed in each well of a 96-well plate, under aseptic conditions. Before 24 h of incubation at 37 °C, 10 µL of bacterial suspension was added to each well, and the plates were covered. The bacterial growth was determined after the measurement of absorbance, at 620 nm, using the Multiskan FC microplate reader (Thermo Scientific, Loughborough, UK). Assays were carried out in triplicate for each tested microorganism [21,22,23].

### 3.8. Antiacetylcholinesterase Assay

The acetylcholinesterase (AChE) inhibitory assay was performed by a modified Ellman method described by Rijo et al. [24]. A sample of the prepared aqueous extract was added to an acetylcholinesterase solution in a buffer of pH 8. After incubation, a solution of acetylcholine and Ellman reagent were added and the absorbance was measured at 405 nm when the reaction reached equilibrium. A control reaction was carried out using the buffer as a replacement for the extract, and it was deemed to have 100% activity. As a positive control, 3 µM tacrine was used.

All assays were performed in triplicate, and the inhibition percentage was determined using the following equations, Equations (2) and (3).
AChE velocity reaction = corrected absorbance/time (min)(2)

Equation (2) determines the AChE velocity reaction of the control (ΔAbs_405nm_/min), which should be in the linear range.
AChE Inhibition % = 100 − [(100 × velocity reaction of inhibitor)/velocity reaction of control](3)

Equation (3) determines the percentage inhibition by the aqueous extract.

### 3.9. Antioxidant Activity

The assessment of the antioxidant properties in terms of reducing capacity and the extract concentration providing 50% of antioxidant activity (IC_50_) was evaluated by the 2,2-diphenyl-1-picrylhydrazyl, DPPH radical method as previously described [24,25,26].

Plant extract (10 μL) was added to a 990 μL solution of DPPH (0.002% in methanol). The mixture was incubated at room temperature for 30 min. The absorbance was measured at 517 nm against a corresponding blank, and the antioxidant activity was calculated as in Equation (4). All assays were carried out in triplicate. The IC_50_ was obtained by plotting the DPPH% radical scavenging activity against the sample concentration.
DPPH% radical scavenging activity = [(A_DPPH_ − A_sample_)/A_DPPH_] × 100(4)

Equation (4) determines the antioxidant activity, where A_DPPH_^∙^ is the absorbance of DPPH against the blank and A_sample_ is the absorbance of the extract, or the control, against the blank.

## 4. Conclusions

*P. ecklonii* Benth. was studied due to its ethnobotanical skin uses, with two formulations containing *P. ecklonii* organic and aqueous extracts. The components RA acid and ParvD were adequately identified and quantified in the *P. ecklonii* extracts studied in this work.

In aqueous solution, RA was able to permeate the skin; however, the same compound in *P. ecklonii* aqueous extract was not detected in the receptor chamber of Franz diffusion cells. ParvD could not be detected mainly due to its high lipophilicity.

The association of the *P. ecklonii* extract with a UV filter substantially elevated its SPF efficacy, giving such a kind of product multifunctional benefits. Besides, no cytotoxicity was found for human keratinocytes. The antibacterial activity and acetylcholinesterase inhibition demonstrated the high potential for the use of this extract as a bioactive ingredient in cosmetics. The microwave aqueous extract of *P. ecklonii* studied herein exhibited antioxidant activity that might be related to its polyphenolic content. Based on the multiple bioactivities of the extract and its active substances, a finished product could be claimed as a multifunctional cosmeceutical with broad skin valuable effects, from UV protection to antiaging action. Since the cutaneous permeation analysis suggested virtually no passage of the major bioactive compound of the extract across the human epidermis, potential superficial topical use of the extract was identified.

## Figures and Tables

**Figure 1 pharmaceuticals-13-00120-f001:**
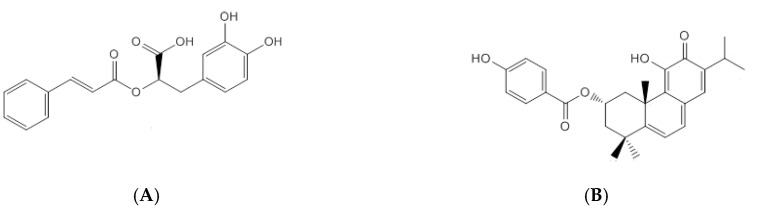
Chemical structures of rosmarinic acid (**A**) and parvifloron D (**B**).

**Figure 2 pharmaceuticals-13-00120-f002:**
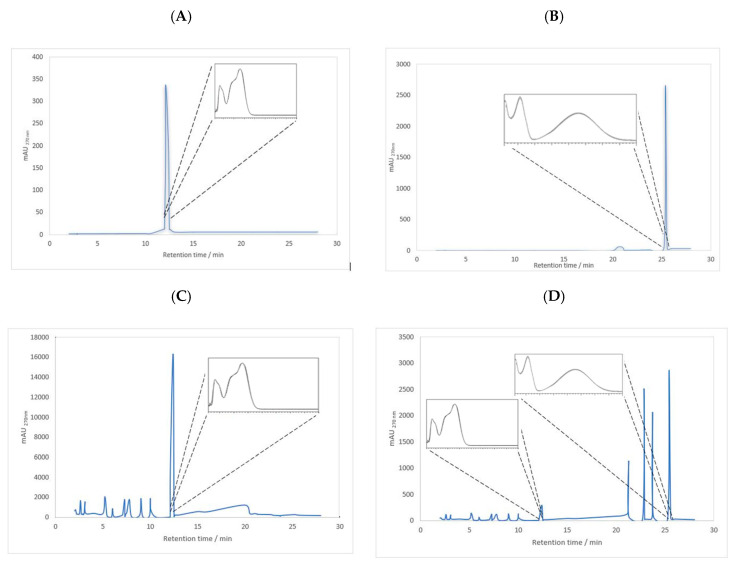
Chromatogram (270 nm) and UV-visible spectrum (210–610 nm) of RA in water (**A**), ParvD in ethanol/PG (70:30, *v*/*v*) (**B**), *P. ecklonii* aqueous extract (**C**), and *P. ecklonii* ethanol/PG (70:30, *v*/*v*) extract (**D**).

**Figure 3 pharmaceuticals-13-00120-f003:**
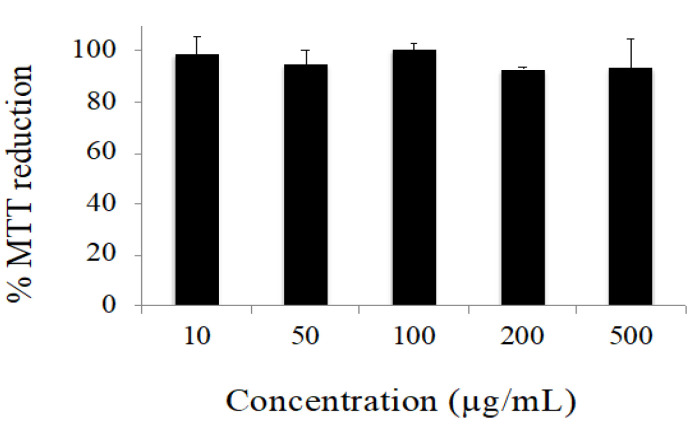
Effect of *P. ecklonii* aqueous extract on the viability of human keratinocytes, as evaluated by the MTT assay. Cells were incubated with increasing concentrations of the extract for 24 h. Results are average values ± SD from two independent experiments, each comprising four replicate cultures.

**Table 1 pharmaceuticals-13-00120-t001:** Initial concentrations of *P. ecklonii* extract, rosmarinic acid, and parvifloron D.

Solutions/Extracts	Concentration/mM	
	RA	ParvD
*P. ecklonii* in water	2.02 ± 0.05	nd
*P. ecklonii* in ethanol/PG (70:30, *v*/*v*)	0.29 ± 0.04	3.13 ± 0.22
RA in water	2.00 ± 0.01	nd
ParvD in ethanol/PG (70:30, *v*/*v*)	nd	3.15 ± 0.03

(nd—not detected).

**Table 2 pharmaceuticals-13-00120-t002:** Permeation data after 48 h.

Receptor Solutions	RA Concentration / µM
*P. ecklonii* in water	nd
*P. ecklonii* in ethanol/PG (70:30, *v*/*v*)	nd
RA in water	80.00 ± 0.01
ParvD in ethanol/PG (70:30, *v*/*v*)	nd

(nd—not detected).

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
