# Peer review of "Assessment of the Potential Skin Application of *Plectranthus ecklonii* Benth."

_pharmaceuticals, 2020, doi:10.3390/ph13060120_

Round 1

Reviewer 1 Report

An interesting study but several aspects need some corrections and better explanation:

  1. Lines 79-88, please rephrase this paragraph, some statements should be transfered to discussion section.
  2. Plants were collected in June 1998 - could such a long shelf life affect the content of bioactive ingredients and their activity? Under what conditions were the plants stored?
  3. Mathematical formulas should be made in the editor, not as pictures, it would increase their readability.
  4. Line 205 - author's name missing.
  5. Lines 303-309 - Since the authors stated that the extracts had antibacterial activity, they should include the results in the form of the size of zones of inhibited growth.  
  6. Table 3 (NMR) - what is the purpose of these results? This is not mentioned in the materials and methods section.

Author Response

Reviewer #1: Comments to the Authors

Comment 1: An interesting study but several aspects need some corrections and better explanation:

Lines 79-88, please rephrase this paragraph, some statements should be transfered to discussion section.

Authors: Thank you for the constructive comment, the paragraph has been modified.

Comment 2: Plants were collected in June 1998 - could such a long shelf life affect the content of bioactive ingredients and their activity? Under what conditions were the plants stored?

Authors: Thank you for the comment. This study was performed in 2013, but the researchers personal lives changed and we start in 2018 to construct the manuscript publish the results. So the plants were collected in June and in 2011-2012 were worked and in 2013 the study was performed, so no problem with the integrity of the plants.

Comment 3: Mathematical formulas should be made in the editor, not as pictures, it would increase their readability.

Authors: Thank you for the suggestion, the formulas were changed and are not as pictures but as equations, as you proposed.

Comment 4: Line 205 - author's name missing.

Authors: Thank you for the constructive comment, the author name was introduced in the text.

Comment 5: Lines 303-309 - Since the authors stated that the extracts had antibacterial activity, they should the results in the form of the size of zones of inhibited growth.

Authors: Thank you for the comment. Indeed the extracts were screen using the well diffusion test but the data are not shown, it was just mention to justify why we continue directly to the microdilution method where the MIC value was determined as 10  μg.mL-1.

Comment 6: Table 3 (NMR) - what is the purpose of these results? This is not mentioned in the materials and methods section.

Authors: Thank you for the constructive comment. This table was removed of the manuscript and was included in the Supplementary Information to justify and confirm the structure of Parvifloron D using NMR spectra data.   

Reviewer 2 Report

Comments to the Editor/ Authors:

The authors of this paper present an interesting study on the Assessment of the potential skin application of Plectranthus ecklonii Benth. Nevertheless, this work might be polished up taking into account the following suggestions:

During the initial checking of the manuscript, we saw that there are few overlaps in your manuscript. That means the highlighted parts (see attached file) is similar to some published works. Please kindly revise/refresh these parts in your manuscript in order to improve the originality of your paper.

Please follow journals authors instructions about manuscript sections format.

*Abstracts   *Introduction *Results *Discussion   * Materials and Methods *Conclusion

  • Abstract:

COMMENT: Well described

  • Introduction:

COMMENT: In the introduction section the effects of polyphenols on cells be strengthened by describing also some benefits from the reference Skenderidis et al. Assessment of the antioxidant and antimutagenic activity of extracts ………..

Materials and Methods:

COMMENT: Generally, well described, in the plant material section the addition of the Latitude data of the plantation is proposed. Please report the DPPH radical as DPPH· in all manuscript.

  • Results:

COMMENT: Please improve the quality of figures in order to be more legible.

  • Discussion:

COMMENT:  The presented results are not discussed accordingly. The discussion is poorly written and should be revised (in a separate discussion section according to journal instructions) and re-written well and compare your results with other papers.

To conclude, reported data are sufficiently presented and commented and the results support sufficiently the authors conclusion. Therefore, I think that this paper is suitable for publication after the correction of the above observations.

Author Response

Reviewer #2: Comments to the Authors

The authors of this paper present an interesting study on the Assessment of the potential skin application of Plectranthus ecklonii Benth. Nevertheless, this work might be polished up taking into account the following suggestions:

Comment 1: During the initial checking of the manuscript, we saw that there are few overlaps in your manuscript. That means the highlighted parts (see attached file) is similar to some published works. Please kindly revise/refresh these parts in your manuscript in order to improve the originality of your paper.

Authors: Thank you for the correction. Lines 30, 31, 51, 52, 53, 54, 57, 58, 64, 73, 94, 95, 114-119, 123-130 ,131-134, 159-166, 169-175, 184-194, 197-201, 206-209, 219-226, 324 were rewritten and are highlighted in the manuscript.

Comment 2: Please follow journalsn authors instructions about manuscript sections format.

*Abstracts   *Introduction *Results *Discussion   * Materials and Methods *Conclusion

Authors: Thank you for the constructive comment. As suggested he order of manuscript sections has been adjusted.

Comment 3: Abstract:

COMMENT: Well described

Introduction:

In the introduction section the effects of polyphenols on cells be strengthened by describing also some benefits from the reference Skenderidis et al. Assessment of the antioxidant and antimutagenic activity of extracts ………..

Authors: Thank you for careful and constructive observations. This reference was introduced in the section of antioxidant activity.

Comment 4: Materials and Methods:

Generally, well described, in the plant material section the addition of the Latitude data of the plantation is proposed. Please report the DPPH radical as DPPH· in all manuscript.

Authors: Thank you for the constructive comment. The information (Latitude: 38°46'N; Longitude: 9°08'W; Altitude: 100 m) was included in the plant material section. Considering the antioxidant activity, the DPPH radical was replaced by DPPH· in all the manuscript.

Comment 5: Results: Please improve the quality of figures in order to be more legible.

Authors: Thank you for the suggestion. The figures was improved, we hope it is suitable for this review.

Comment 6: Discussion: The presented results are not discussed accordingly. The discussion is poorly written and should be revised (in a separate discussion section according to journal instructions) and re-written well and compare your results with other papers.

To conclude, reported data are sufficiently presented and commented and the results support sufficiently the authors conclusion. Therefore, I think that this paper is suitable for publication after the correction of the above observations.

Authors: Thank you for the comment, the discussion was improved and is highlighted in the manuscript.

Reviewer 3 Report

The reviewed manuscript provides information on the use of Plectranthus ecklonii extract in human skin formulations, with the aim to demonstrate a potential cosmetic and pharmacological action. Since this article may give an additional economic and pharmaceutical values to a plant that actually is little used for the cosmetic formulation, the topic may be interesting. Indeed, this plant showed in the past interesting biological and cosmetic application, especially in African folk medicine.

However, even if the obtained data show potentially interesting results, the way in which they are described and discussed in the manuscript follow in the impossibility of their publication on “Pharmaceuticals”.

What easily stands out after a first quick reading of the manuscript, is the misuse of the references. The authors used only 22 scientific articles, of which 19 are self-citations. Moreover, the introduction is not well-written, very general, and it completely lack proper references. Indeed, several statements were made without any bibliographic citation. In particular, the statements at line 43, 46, 48, 51, 60-61, 72, 74 and 76. My doubt is that the authors are not interested in making references, if not their own articles.

Moreover, several errors in English writing are present all over the text (i.e. line 82).

At LINE 94-95 authors wrote the plant identification was made by Enrico S. Martins. Who is he? Is he a botanist? Please, provide the academic title, as well as any other qualification that could suggest to readers that Mr. Martins is the most appropriate person for the recognition of this plant.

Section 2.5: Authors described the HPLC method, using three different solvent (A, B and D). I can understand that the authors placed the solvents on the line A, B and D during the analysis but, since it is not a technically relevant fact, replace the letter D with C. Moreover, in this section is not explained the methods used for the quantification neither of RA nor of parvifloron D (Did they maybe use an external calibration curve of RA and parvifloron D?). In addition, authors declared that RA acid was not detected during the permeation test, or that the extract presented only RA. How can they state it with absolute certainty without calculate a LOD and LOQ? Finally, a part LOD and LOQ, miss any information regarding matrix effects, linearity and precision of the analytical method.

Since rosmaric acid appears several times in the text, authors should always use the acronym (RA).

Equation: all the equations reported in the article should be written using the appropriate Microsoft Word tool. They are few clear. Furthermore, each punctual abbreviation of the equation should be explained in the text. Above all, they should explain it the first time that it appears in the text (in the introduction), and they should avoid sometimes writing the name in full, and other times as acronym.

The results and discussion paragraph is actually only a mere description of the results. It is insufficient, with few references to previously published data. Some paragraphs (cytotoxicity, microbiological and antioxidant activity) are not even discussed.

Table 1 reports no sense data (what means 0.00 ± 0.01?)

Figure 1: the caption of this figure is insufficient and incomplete.

Author Response

Reviewer #3: Comments to the Authors

Comment 1: The reviewed manuscript provides information on the use of Plectranthus ecklonii extract in human skin formulations, with the aim to demonstrate a potential cosmetic and pharmacological action. Since this article may give an additional economic and pharmaceutical values to a plant that actually is little used for the cosmetic formulation, the topic may be interesting. Indeed, this plant showed in the past interesting biological and cosmetic application, especially in African folk medicine.

However, even if the obtained data show potentially interesting results, the way in which they are described and discussed in the manuscript follow in the impossibility of their publication on “Pharmaceuticals”.

What easily stands out after a first quick reading of the manuscript, is the misuse of the references. The authors used only 22 scientific articles, of which 19 are self-citations. Moreover, the introduction is not well-written, very general, and it completely lack proper references. Indeed, several statements were made without any bibliographic citation. In particular, the statements at line 43, 46, 48, 51, 60-61, 72, 74 and 76. My doubt is that the authors are not interested in making references, if not their own articles.

Authors: Thank you for the constructive comment. We do not want to misuse the references and only cite our papers so we change the manuscript and in lines 43, 46, and 48, was introduced a new reference. In lines 61 and 62 we introduced a reference and in lines 71-78 we cite new references:

7-Chen, J., Hammell, D.C., Spry, M., D'Orazio, J.A. and Stinchcomb, A.L. (2009). In vitro skin diffusion study of pure forskolin versus a forskolin-containing Plectranthus barbatus root extract. Journal of Natural Products, 72, 769-771.

8-Nyila, M.A., Leonard, C.M., Hussein, A.A. and Namrita, L.N.N. (2009). Bioactivities of Plectranthus ecklonii Constituents. Natural Product Communications. 4, 1177-1180.

Comment 2: Moreover, several errors in English writing are present all over the text (i.e. line 82).

Authors: We appreciate your contribution to the improvement of our manuscript. The paragraph lines 79-88 were modified and we correct all the manuscript to improve and correct all the errors in english.

Comment 3: At LINE 94-95 authors wrote the plant identification was made by Enrico S. Martins. Who is he? Is he a botanist? Please, provide the academic title, as well as any other qualification that could suggest to readers that Mr. Martins is the most appropriate person for the recognition of this plant.

Authors: Thank you for your observation. Dr. Enrico S. Martins is retired so the phrase has been modified in the manuscript:

  1. ecklonii Benth. was grown in the Hortum of the Faculty of Pharmacy (Latitude: 38°46'N; Longitude: 9°08'W; Altitude: 100m; Lisbon University, Portugal) from seeds provided by the Herbarium of the National Botanic Garden of Kirstenbosch, South Africa. Entire plants were collected in July 1998, identified, and deposited in the Herbarium of the Instituto de Investigação Científica Tropical, Lisbon (ref. C. Marques S/No. LISC).

Comment 4: Section 2.5: Authors described the HPLC method, using three different solvent (A, B and D). I can understand that the authors placed the solvents on the line A, B and D during the analysis but, since it is not a technically relevant fact, replace the letter D with C. Moreover, in this section is not explained the methods used for the quantification neither of RA nor of parvifloron D (Did they maybe use an external calibration curve of RA and parvifloron D?). In addition, authors declared that RA acid was not detected during the permeation test, or that the extract presented only RA. How can they state it with absolute certainty without calculate a LOD and LOQ? Finally, a part LOD and LOQ, miss any information regarding matrix effects, linearity and precision of the analytical method.

Authors: Thank you for the correction. The described HPLC method was modified and LOQ and LOD values were introduced:

“RA and ParvD were detected and quantified by the HPLC-DAD Agilent Technologies 1200 Infinity Series system with ChemStation software in combination with a LiChrospher® 100, RP-18 (5 µm) Merck column. The samples were analyzed injecting 20 µL and using 1 mL.min-1 flow rate and a gradient composed of solution A (methanol), solution B (acetonitrile) and solution C (0.3% trichloroacetic acid in water) as follows: 0 min, 15% A, 5% B and 80% C; 20 min, 80% A, 10% B, and 10% C and 28 min, 80% A, 10% B and 10% C. The standards methanol solutions were run under the same conditions and the detection was carried out at 270 nm with a diode array detector (DAD). All analyses were performed in triplicate. The limit of detection (LOD) and limit of quantification (LOQ) for RA and ParvD were evaluated.”

Comment 5: Since rosmaric acid appears several times in the text, authors should always use the acronym (RA).

Authors: Thank you for the constructive comment and rosmaric acid was replaced by RA in all the manuscript.

Comment 6: Equation: all the equations reported in the article should be written using the appropriate Microsoft Word tool. They are few clear. Furthermore, each punctual abbreviation of the equation should be explained in the text. Above all, they should explain it the first time that it appears in the text (in the introduction), and they should avoid sometimes writing the name in full, and other times as acronym.

Authors: Thank you for the observation. All equations were written using the appropriate Microsoft word tool. The abbreviations of the equations were explained.

Comment 7: The results and discussion paragraph is actually only a mere description of the results. It is insufficient, with few references to previously published data. Some paragraphs (cytotoxicity, microbiological and antioxidant activity) are not even discussed.

Authors: Thank you for the correction, we improve the discussion for all the results presented.

Comment 8: Table 1 reports no sense data (what means 0.00 ± 0.01?)

Authors: Thank you for the correction, we followed your suggestion and changed the content of Table 1.

Comment 9: Figure 1: the caption of this figure is insufficient and incomplete

Authors: Grateful for the correction. The caption of figure 1 was modified.

Round 2

Reviewer 1 Report

The authors improved the manuscript following the suggestions. 

Reviewer 3 Report

The authors followed all the recommendations suggested in my previous report. Now, the paper can be considered for a potential publication in Pharmaceuticals.